# Navigating the Landscape of Personalized Medicine: The Relevance of ChatGPT, BingChat, and Bard AI in Nephrology Literature Searches

**DOI:** 10.3390/jpm13101457

**Published:** 2023-09-30

**Authors:** Noppawit Aiumtrakul, Charat Thongprayoon, Supawadee Suppadungsuk, Pajaree Krisanapan, Jing Miao, Fawad Qureshi, Wisit Cheungpasitporn

**Affiliations:** 1Department of Medicine, John A. Burns School of Medicine, University of Hawaii, Honolulu, HI 96813, USA; noppawit@hawaii.edu; 2Division of Nephrology and Hypertension, Department of Medicine, Mayo Clinic, Rochester, MN 55905, USA; supawadee.sup@mahidol.ac.th (S.S.); pajareek@tu.ac.th (P.K.); miao.jing@mayo.edu (J.M.); qureshi.fawad@mayo.edu (F.Q.); cheungpasitporn.wisit@mayo.edu (W.C.); 3Chakri Naruebodindra Medical Institute, Faculty of Medicine Ramathibodi Hospital, Mahidol University, Samut Prakan 10540, Thailand; 4Department of Internal Medicine, Faculty of Medicine, Thammasat University, Pathum Thani 12120, Thailand

**Keywords:** literature review, nephrology references, ChatGPT, Bing Chat, Bard AI, accuracy, personalized medicine, precision medicine

## Abstract

Background and Objectives: Literature reviews are foundational to understanding medical evidence. With AI tools like ChatGPT, Bing Chat and Bard AI emerging as potential aids in this domain, this study aimed to individually assess their citation accuracy within Nephrology, comparing their performance in providing precise. Materials and Methods: We generated the prompt to solicit 20 references in Vancouver style in each 12 Nephrology topics, using ChatGPT, Bing Chat and Bard. We verified the existence and accuracy of the provided references using PubMed, Google Scholar, and Web of Science. We categorized the validity of the references from the AI chatbot into (1) incomplete, (2) fabricated, (3) inaccurate, and (4) accurate. Results: A total of 199 (83%), 158 (66%) and 112 (47%) unique references were provided from ChatGPT, Bing Chat and Bard, respectively. ChatGPT provided 76 (38%) accurate, 82 (41%) inaccurate, 32 (16%) fabricated and 9 (5%) incomplete references. Bing Chat provided 47 (30%) accurate, 77 (49%) inaccurate, 21 (13%) fabricated and 13 (8%) incomplete references. In contrast, Bard provided 3 (3%) accurate, 26 (23%) inaccurate, 71 (63%) fabricated and 12 (11%) incomplete references. The most common error type across platforms was incorrect DOIs. Conclusions: In the field of medicine, the necessity for faultless adherence to research integrity is highlighted, asserting that even small errors cannot be tolerated. The outcomes of this investigation draw attention to inconsistent citation accuracy across the different AI tools evaluated. Despite some promising results, the discrepancies identified call for a cautious and rigorous vetting of AI-sourced references in medicine. Such chatbots, before becoming standard tools, need substantial refinements to assure unwavering precision in their outputs.

## 1. Introduction

The digital era has brought about transformative changes in various aspects of our lives, with the medical field being no exception [1,2]. Within the vast expanse of medical literature, scholars, clinicians, and medical professionals rely heavily on evidence-based studies to formulate decisions, guidelines, and recommendations for patients [3]. Literature reviews play an instrumental role in this process, often serving as the cornerstone to understanding the ever-expanding universe of medical evidence [4]. However, while the volume of information is expanding, so is the need for efficient tools to extract relevant knowledge.

The growing number of published articles has led to a substantial increase in the references that physicians and researchers must stay updated with. As of 2020, there were over 30 million articles indexed in PubMed alone, with an estimated addition of a million entries each year [5]. This exponential growth makes the task of manually extracting, comparing, and verifying references not only laborious but also prone to human errors [6]. In this context, AI-powered platforms are emerging as potential aides for literature reviews [7]. Contemporary innovations have introduced platforms such as ChatGPT [8], Bing Chat [9] and Bard AI [10]. These tools are not just digital cataloging systems but smart engines that claim to understand and retrieve precise information. The allure of such platforms lies in their ability to rapidly sift through vast data sets, potentially offering precise references that would take humans considerably longer to extract [11,12].

The emergence of ChatGPT, a creation of OpenAI, introduces promising prospects spanning a variety of domains, with a pronounced emphasis on the enrichment of healthcare education [13]. This AI framework not only highlights its advanced acumen in information retrieval but also adeptly addresses syntactical inaccuracies, thereby serving as a valuable resource for literature evaluations and the composition of scholarly manuscripts [8]. In a parallel vein, Bing Chat, a product of Microsoft, emerges as an AI-driven conversational agent capable of engendering inventive and novel content, spanning the spectrum from poetic compositions and narratives to code snippets, essays, musical compositions, satirical renditions of celebrities, and visual representations [9]. Akin to its counterparts, Bard AI, the brainchild of Google, assumes its stance as a formidable entity within the domain of AI models, having undergone rigorous training on an expansive corpus encompassing textual and code-oriented knowledge culled from diverse sources, including literary works and academic articles [10]. The transformative potential of these technological tools in revolutionizing the paradigm of information retrieval is evident; however, their precision, particularly in terms of adhering to meticulous citation protocols, remains subject to meticulous examination.

The accuracy of citations within scholarly discourse is far from being a mere ritualistic practice; rather, it holds a pivotal role. These references provide a conduit for readers to retrace the steps back to original sources, thereby ensuring the veracity of derived conclusions and recommendations firmly anchored in authentic research endeavors. The presence of even minute inaccuracies within references can cast a shadow of doubt over the entirety of a scholarly paper, thereby undermining both its credibility and the integrity of the author [14]. This holds critical importance, especially in specialized fields such as Nephrology, where medical treatments have far-reaching effects on patient health and general well-being, including risks such as kidney failure or allograft rejection or failure. A single incorrect reference carries the risk of initiating misunderstandings, which could eventually lead to less than ideal or even harmful clinical decisions. The present investigation, therefore, is conceived with the overarching aim of unraveling the precision exhibited by these emergent AI entities in the realm of citations, particularly within the highly specialized terrain of Nephrology.

The purpose of this study is to assess the citation accuracy of AI models including ChatGPT (versions 3.5 and 4.0) [15], Bing Chat, and Bard AI in retrieving and validating references for academic research in nephrology.

## 2. Materials and Methods

### 2.1. Search Strategy and Criteria

We used three distinct AI chatbots to perform literature searches in nephrology, including (1) ChatGPT, (2) Bing Chat, and (3) Bard AI. To ensure the comparability of these chatbots, we standardized the criteria for evaluating their performance based on search results’ relevance, comprehensiveness, and the timeliness of the articles retrieved. ChatGPT is a large language model developed by OpenAI and integrates both GPT-3.5 [16] and GPT-4.0 models [15] that comprehend and generate human-like responses through text. Bing Chat is powered by GPT-4.0 and incorporated into the Microsoft Edge browser, which has another capability to generate images and innovative content [9]. Bard AI, a robust Large Language Model (LLM) developed by Google based on Pathways Language Model 2 (PaLM2) and trained on an expansive collection of text and code that exhibits creative content design.

On 1 August 2023, we generated the prompt to ask AI chatbots to provide 20 references in Vancouver style, a commonly used citation style in academic writing in each Nephrology topics; The topics were chosen to reflect a comprehensive understanding of the field and were pre-determined through a review of the most common nephrology subjects discussed in existing literature. (1) general nephrology, (2) glomerular disease, (3) hypertension, (4) acute kidney injury, (5) chronic kidney disease, (6) end-stage kidney disease, (7) electrolyte disorders, (8) acid-base disturbances, (9) kidney stones, (10) hemodialysis, (11) peritoneal dialysis and (12) kidney transplantation. There was slight difference between the prompts used for each individual AI chatbot as we modified the prompt to optimize their responses; “please provide 20 references in Vancouver style and links of the most updated literatures regarding (Nephrology topic)” for ChatGPT, “provide Vancouver references with DOI of 20 articles on (Nephrology topic)” for Bing Chat, and “20 updated references regarding (Nephrology topic) in Vancouver style” for Bard AI. We used only GPT-3.5 model for ChatGPT because GPT-4.0 was unable to provide actual references but it solely provided examples of references despite our attempt to modify the prompt used. We documented five key components of provided references: (1) author name, (2) publication title, (3) journal title, (4) publication year or issue and (5) digital object identifier (DOI). This categorization was performed to ensure the verification process adhered to a uniform criterion for all three chatbots, facilitating a balanced assessment.

We verified the existence and accuracy of the references using several medical literature databases. The databases were selected based on their reputability and coverage in the field of nephrology. We initially used the provided DOI to search for its corresponding references in PubMed [5], the widely recognized database in biomedical literatures. If we could not find the reference in PubMed or we had incomplete or missing DOI, we used Google Scholar [17] or Web of Science [18] as additional databases for comprehensive search. We used University of Hawaii library website [19] and google search to check references of textbook or book chapter.

We categorized the validity of the provided references from the AI chatbot into following groups; (1) incomplete, (2) fabricated, (3) inaccurate, and (4) accurate. These categories were defined to allow for the precise characterization of the search results, which is crucial for determining the reliability and utility of AI-generated references. Reference was defined as incomplete when the provided reference information was inadequate to verify its existence in aforementioned medical databases. Reference was defined as fabricated when we could not find the reference in the database. Reference was defined as existing but inaccurate when we could identify the reference in the database but at least one of five reference components were incorrect. Reference was defined as existing and accurate when we could identify the reference in the database and all of five reference components were correct. A flow diagram of the research methodology was Illustrated in Figure 1 and the example of an assessment of references was illustrated in Figure 2.

To assess both the magnitude and directionality of linear relationships among different performance indicators of the chatbots, we computed Pearson correlation coefficients. The Pearson correlation method was chosen for its sensitivity to linear associations, making it well-suited for our dataset, which we ascertained met the assumptions of linearity, normality, and homoscedasticity. In these matrices, individual cells contain the computed Pearson coefficients, which are bounded between −1 and 1. The closer a coefficient is to 1, the stronger the positive linear relationship, indicating that an increase in one performance metric is likely paralleled by an increase in another. In contrast, a coefficient value nearing −1 reveals a strong negative relationship, meaning that a rise in one metric typically results in a decline in another. These boundaries are strict and allow for nuanced interpretation: a value of exactly 1 or −1 would signify a perfect linear relationship, positive or negative respectively, although such a result is exceedingly rare in practical applications. Coefficients approximating zero signify weak or negligible linear relationships, implying that changes in one variable are not systematically accompanied by changes in another. The use of Pearson correlation analysis in this context is instrumental for pinpointing specific performance metrics that may be most amenable to enhancements, thereby aiding in targeted optimization of chatbot functionalities.

### 2.2. Statistical Analysis

We reviewed the references and excluded duplicated reference from the same AI chatbot before analysis. We presented the validity of the provided references as counts with percentages and compared among AI chatbots using Chi-squared test. *p*-value < 0.05 was considered statistically significant. IBM SPSS statistics version 26 was used for all statistical analyses.

The calculations of Pearson correlation coefficients involved generating correlation matrices, executed with Python’s Seaborn library—a tool that efficiently interfaces with Pandas for data structuring and Matplotlib for graphical output.

## 3. Results

Although each AI chatbots was expected to provide 240 references (20 references in each of 12 nephrology topics), 41 (17%), 65 (29%) and 39 (26%) references provided by ChatGPT, Bing Chat and Bard, respectively, were found as duplicated references, while 17 (7%) references from Bing Chat and 89 (37%) references from Bard were absent.

A total of 199 (83%), 158 (66%) and 112 (47%) unique references were provided from ChatGPT, Bing Chat and Bard, respectively. ChatGPT provided 76 (38%) accurate, 82 (41%) inaccurate, 32 (16%) fabricated and 9 (5%) incomplete references. Bing Chat provided 47 (30%) accurate, 77 (49%) inaccurate, 21 (13%) fabricated and 13 (8%) incomplete references. In contrast, Bard provided 3 (3%) accurate, 26 (23%) inaccurate, 71 (63%) fabricated and 12 (11%) incomplete references. The proportion of existing references were similar between ChatGPT and Bing Chat, but ChatGPT provided higher proportion of accurate references. Bard had the highest proportion of fabricated and incomplete references. There were statistically significant differences in proportion of accurate, inaccurate and fabricated references between ChatGPT and Bard, and between Bing Chat and Bard. However, the validity between ChatGPT and Bing Chat did not significantly differ (Table 1 and Figure 3).

When we assessed the reason for inaccurate references, DOI was the most common inaccurate component in references provided by ChatGPT and Bing Chat, followed by author name, publication year/issue, journal title, and reference title. In contrast, author name was the most common reason inaccurate component in references provided by Bard, followed by DOI, publication year/issue, journal title, and reference title. Significant differences of all inaccuracy domains were found between ChatGPT and Bard, and between Bing Chat and Bard, while ChatGPT and Bing Chat were not statistically different (Table 2 and Figure 4 and Figure 5).

### 3.1. Correlation Analysis

#### 3.1.1. Validity Metrics

Accurate vs. Inaccurate: A negative correlation of −0.92 suggests that as the accuracy of a chatbot increases, the inaccuracy decreases.

Inaccurate vs. Incomplete: A high positive correlation (0.99) is observed, indicating that chatbots that inaccurate information are also likely to provide incomplete answers (Figure 6).

#### 3.1.2. Incorrect Information Metrics

Missing DOI vs. Wrong DOI: A strong negative correlation of −0.75 implies that chatbots that often miss DOIs are less likely to provide wrong DOIs.

Wrong Author vs. Wrong Journal/Book: A negative correlation of −0.58 suggests that if a chatbot frequently gets the author wrong, it is less likely to get the journal or book wrong (Figure 7).

#### 3.1.3. Missed and Duplicate Metrics

Missed vs. Duplicate: A strong negative correlation of -1 suggests that chatbots that miss information are unlikely to produce duplicate information (Figure 8).

## 4. Discussion

The advancement of AI within the medical field has led to substantial transformations [20,21], including assisting in the specialized diet supports [22,23], prevention of potential allergic reactions [24,25], detection of prescription errors [26], extraction of drug interactions from the literature [27], and particularly concerning literature reviews [28,29]. As the production of medical evidence continues to grow at an accelerated rate, the need for effective tools to sift through and analyze pertinent information has become critically important. In response to this need, AI-powered platforms such as ChatGPT, Bing Chat, and Bard AI have emerged as potential aids in literature reviews [11,12,29]. The findings of this study, however, demonstrate varying degrees of reliability and validity exhibited by the three generative AI chatbots, namely ChatGPT-3.5, Bing Chat and Bard, when tasked with providing references pertaining to Nephrology subjects. It is important to recognize that the utilization of AI chatbots for generating dependable and valid references is accompanied by certain limitations and challenges, encompassing concerns such as the generation of fabricated, inaccurate and incomplete references.

The study outcomes delineate distinctive patterns regarding citation accuracy within the purview of AI tools. ChatGPT emerges with the highest precision at 38%, emblematic of its adherence to established citation protocols. This, however, coexists with a notable proportion of erroneous references at 41%. In contrast, Bing Chat demonstrates an alternative pattern, characterized by a preponderance of inaccurate references (49%) alongside a relatively diminished occurrence of entirely accurate references (30%). However, the validity of ChatGPT and Bing Chat in providing Nephrology references were not significantly different. Bard AI, conversely, exhibits the highest incidence of fabricated references (63%) and incomplete references (11%), suggesting an avenue for enhancement in its reference generation mechanism. It is pertinent to underscore that the discordances identified, including the misallocation of DOIs, underscore the criticality of scrupulous attention to minutiae within the medical realm, where even minor inaccuracies bear substantial consequences.

The study’s findings underscore the heterogeneity in citation accuracy among the evaluated AI tools. While each tool showcased certain strengths, such as ChatGPT’s higher accuracy rate, the discrepancies identified emphasize the need for careful and rigorous vetting of AI-sourced references in the medical field. The precision and authenticity of references hold critical significance, especially considering the potential consequences of medical decisions that rely on these sources. The substantial discrepancies observed in citation accuracy among these AI chatbots reveal that they may not consistently meet the rigorous standards demanded by the medical realm. ChatGPT’s relatively high accuracy, even with its considerable inaccuracies, signifies a degree of potential utility, but the prevalence of incorrect and fabricated references remains a pressing concern.

In terms of optimizing chatbot performance for specific needs, several strategic considerations emerge from the data. If accuracy is of utmost importance, ChatGPT-3.5 stands out as the most reliable choice, although it would benefit from targeted improvements in areas like DOI accuracy. BingChat, on the other hand, offers a different set of trade-offs: it generates less fabricated information but is more prone to inaccuracies. Therefore, it could serve as a viable option where the fabrication of data is a primary concern. The analysis strongly suggests avoiding the experimental version of Bard as of 1 August 2023, for tasks requiring high reliability, given its alarming rates of both fabricated and inaccurate information. Additionally, each chatbot has identifiable weak areas—ChatGPT-3.5 with incorrect DOIs and BingChat with missing DOIs, for instance. These weaknesses could be mitigated through secondary verification systems. When it comes to handling duplicates and missed responses, ChatGPT-3.5 offers the most favorable profile, despite a 17.08% duplication rate. For those prioritizing quality over quantity, the data indicate that focusing on further optimizing ChatGPT-3.5 is the most effective approach. Advanced methods, such as machine learning algorithms, could be employed to refine its performance in specific areas like DOI accuracy. Finally, for those open to unconventional strategies, exploring ensemble methods that combine the strengths of different chatbots could be a worthwhile avenue for research. While speculative, such an approach could result in a more robust and versatile chatbot solution.

Medical research is a cornerstone of evidence-based practice, where even minor errors can have profound consequences on patient care, clinical decisions, and scientific advancement [30]. The findings of this study serve as a warning against premature reliance on AI-generated references in the medical domain. The presence of inaccuracies, fabrications, and incomplete references is untenable, as these undermine the integrity of scholarly work and compromise the trust placed in medical research. Consequently, policies and guidelines need to be developed to ensure the responsible and ethical integration of AI tools in medical research processes. These policies should emphasize the need for rigorous validation and vetting of AI-generated references before their incorporation into clinical decision-making or research publications.

The correlation analysis revealed several noteworthy aspects that could significantly inform strategies for optimizing chatbot performance. First, there’s a clear trade-off in validity; a chatbot that excels in accuracy tends to produce less inaccurate information. However, such bots might still fabricate or provide incomplete information, necessitating caution. Second, specific risk areas were identified. For instance, chatbots that frequently miss DOIs are less prone to providing incorrect DOIs, and vice versa. This insight could be invaluable for implementing targeted validation strategies. Moreover, the correlation between missed and duplicated information suggests another layer of complexity. While chatbots that frequently miss information are unlikely to produce duplicates, the correlation is strong, thus requiring strategic monitoring. This monitoring could focus on specific error types that a chatbot is prone to making. For example, a chatbot that often produces incorrect information may also be susceptible to delivering incomplete responses. Lastly, the correlations offer avenues for customization and fine-tuning of chatbot behavior. Knowing a bot’s strengths and weaknesses in particular areas enables the implementation of specialized validation or correction systems. As an example, if a chatbot is generally accurate but frequently errs in DOI references, a secondary validation system could be introduced specifically to check and correct DOI information. This approach allows for the leveraging of the chatbot’s strengths while mitigating its weaknesses.

The integration of AI tools in the medical field introduces complex ethical considerations [28,29,31,32,33]. As evidenced by this study, the inaccuracies and fabrications in AI-generated references could potentially lead to misinterpretation, misinformation, and misguided medical decisions. Therefore, it is imperative to establish robust ethical frameworks and policy guidelines that guide the responsible use of AI chatbots in generating references for medical research. Such policies should prioritize patient safety, research integrity, and the advancement of medical knowledge. To address these concerns, regulatory bodies and professional organizations should collaborate to develop guidelines that mandate thorough validation and scrutiny of AI-generated references before they are incorporated into research papers, clinical guidelines, or medical recommendations. These guidelines could stipulate the necessity of human oversight, validation by domain experts, and cross-referencing with established databases. Furthermore, ethical considerations should extend to the transparency of AI-generated content. Users should be informed when references are AI-generated, allowing them to assess the reliability and credibility of the sources. This transparency aligns with the principles of informed consent and empowers readers to make informed judgments about the validity of the information presented.

## 5. Limitations

This study bears several limitations that warrant acknowledgment.
AI platforms: Our assessment exclusively focused solely on ChatGPT (GPT-3.5 and GPT-4.0), Bing Chat, and Bard AI, excluding other emerging AI platforms that may exhibit distinct citation accuracy profiles.Lack of clinical implications: We did not explore the downstream impact of reference inaccuracies on downstream research, clinical decision-making, or patient outcomes, which could provide crucial insights into the practical implications of AI-generated references in the medical domain.Limited citation assessment: While the study accounted for discrepancies in citation elements such as DOIs and author names, we did not investigate potential errors in other bibliographic elements, such as the accuracy of the Vancouver format or page ranges. This omission could underestimate the full scope of inaccuracies present in AI-generated references.Variability due to updates: The AI models used in this study are subject to updates and modifications. The investigation was conducted with specific versions of AI models, and as these models undergo continuous refinement, their citation accuracy may evolve.Scope: The study’s sample size of Nephrology topics and AI-generated references might not fully capture the breadth of medical literature or the complexity of citation accuracy in other medical specialties. The study’s exclusive focus on AI chatbots limits the exploration of potential variations in citation accuracy among different AI-powered tools, such as summarization algorithms or natural language processing applications.Validity of databases: The assessment of AI-generated references relied on cross-referencing with established databases, assuming the accuracy of these databases. Any errors or discrepancies present in the reference databases could influence the study’s findings and conclusions.Chatbot Extensions and Web Search: As the technological landscape evolves, chatbots are increasingly being equipped with the ability to integrate extensions and external resources, including web search functions. While this feature augments the utility of chatbots, it simultaneously introduces another layer of complexity in terms of citation accuracy and source validation. There is an imperative for future studies to critically evaluate the accuracy and reliability of references generated through these additional features.

## 6. Conclusions

This study underscores the foundational significance of unwavering research fidelity within the intricate domain of Nephrology. While the potential of AI tools for streamlining literature reviews is evident, the identified discrepancies call for a cautious and meticulous approach in their utilization. The medical community’s commitment to precision demands that even minor inaccuracies remain unacceptable. As the potential for AI tools to revolutionize medical research and practice persists, it is essential to refine and fortify these chatbots before they can be confidently embraced as standard tools.

## Figures and Tables

**Figure 1 jpm-13-01457-f001:**
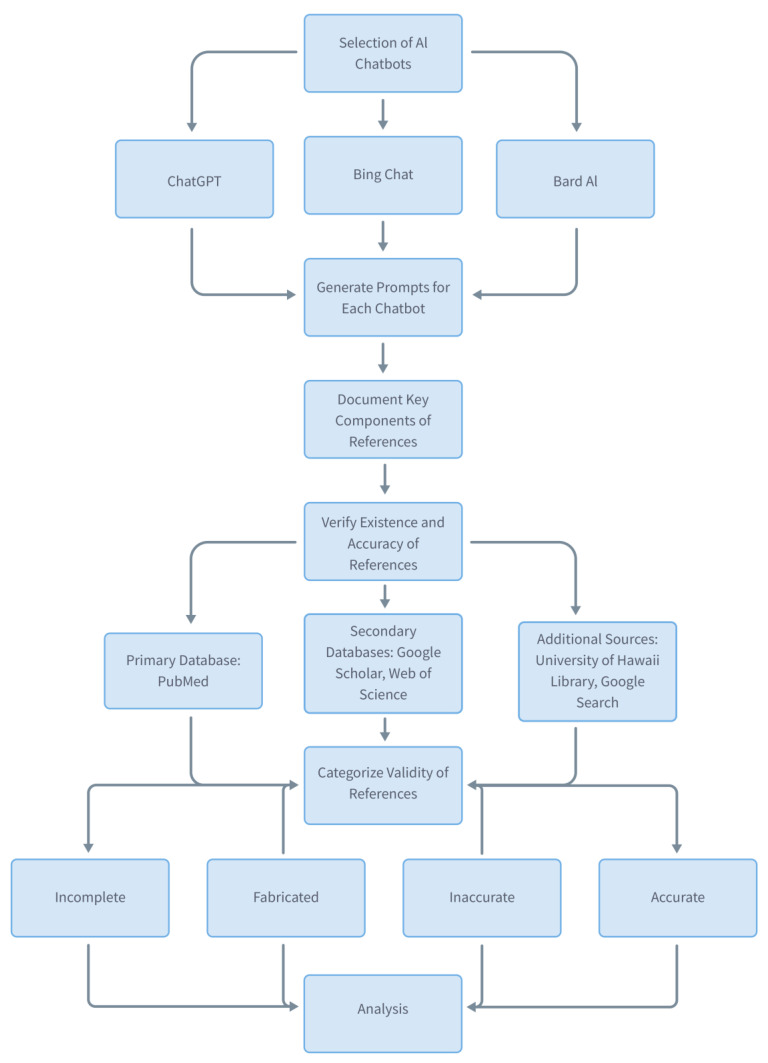
Flow diagram of using AI chatbots for literature search and assessment of validity.

**Figure 2 jpm-13-01457-f002:**
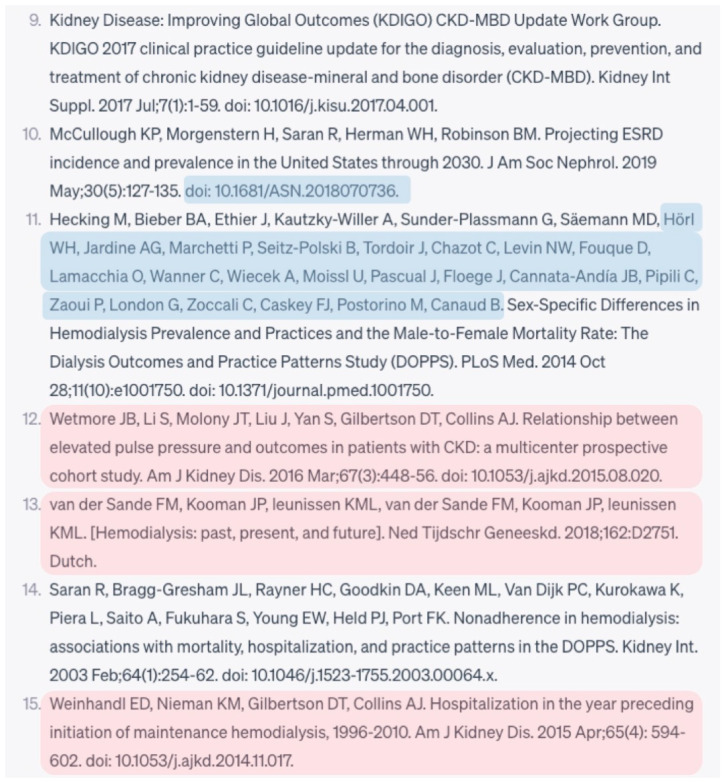
Demonstration of citation assessment regarding hemodialysis topic. The non-highlighted texts represent accurate parts. The blue highlights represent incorrect parts. The pink highlights indicate fabricated references.

**Figure 3 jpm-13-01457-f003:**
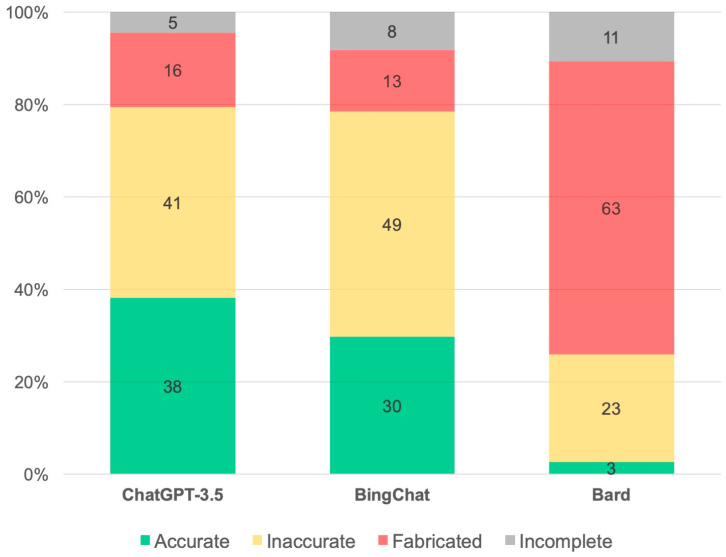
Comparison of validity in providing literature between ChatGPT, Bing Chat and Bard AI.

**Figure 4 jpm-13-01457-f004:**
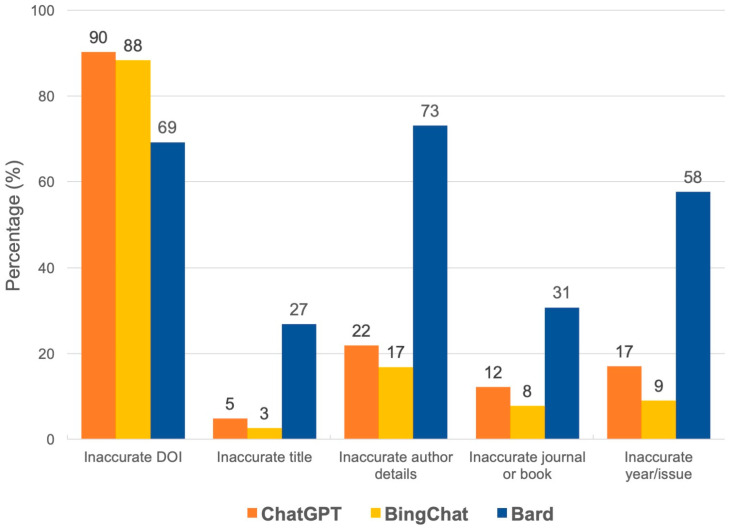
Percentages of each type of inaccuracy among the inaccurate references of ChatGPT, Bing Chat and Bard AI.

**Figure 5 jpm-13-01457-f005:**
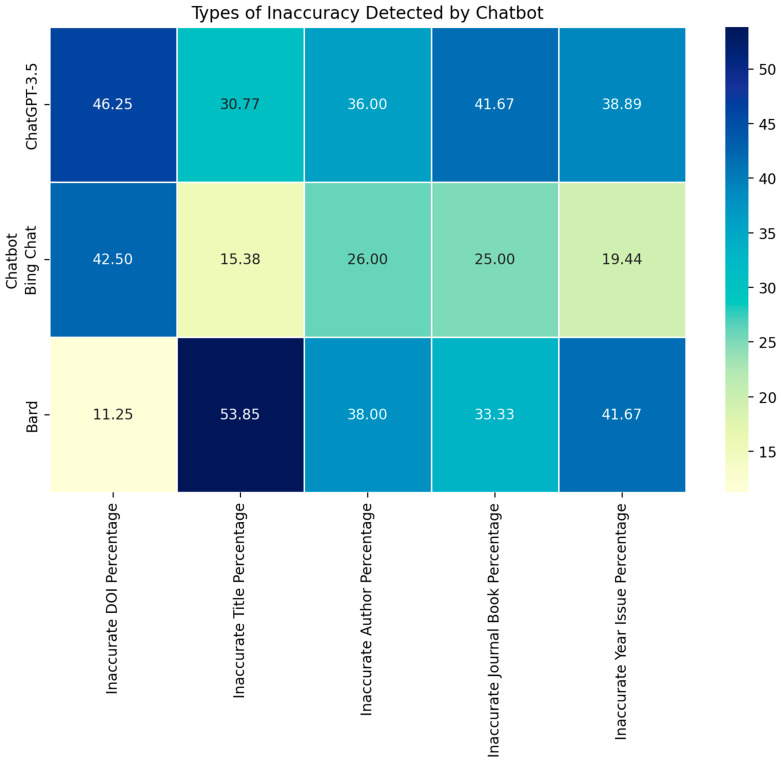
Heatmap visualizing the types of inaccuracies detected among the references provided by each chatbot. The rows represent the chatbots: ChatGPT-3.5, Bing Chat, and Bard. The columns represent the types of inaccuracies: DOI, Title, Author, Journal/Book, and Year/Issue. The intensity of the color indicates the magnitude of the inaccuracy, with darker shades representing higher percentages.

**Figure 6 jpm-13-01457-f006:**
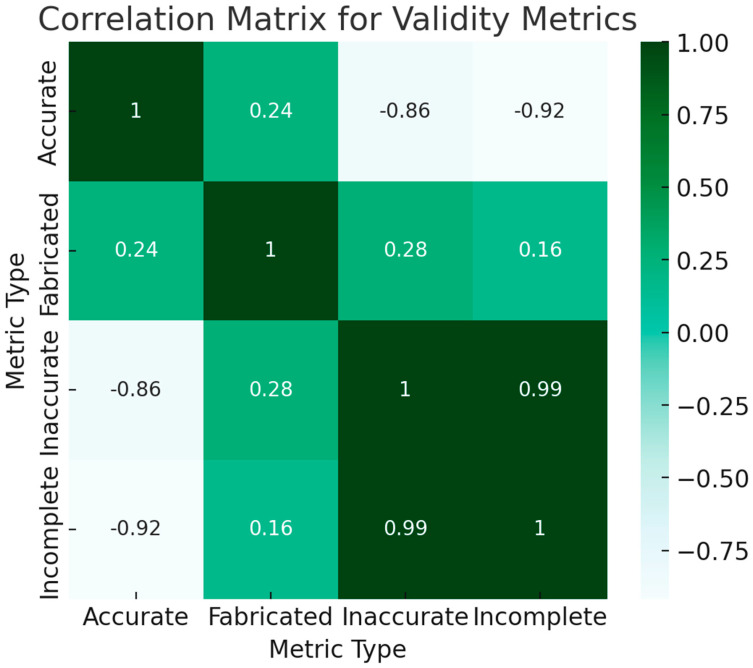
Correlation Matrix for Validity Metrics.

**Figure 7 jpm-13-01457-f007:**
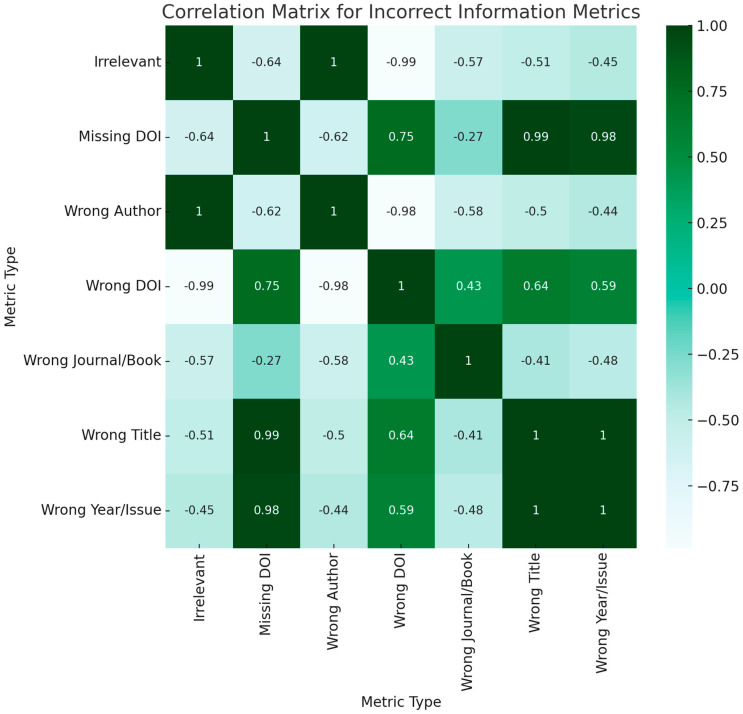
Correlation Matrix for Incorrect Information Metrics.

**Figure 8 jpm-13-01457-f008:**
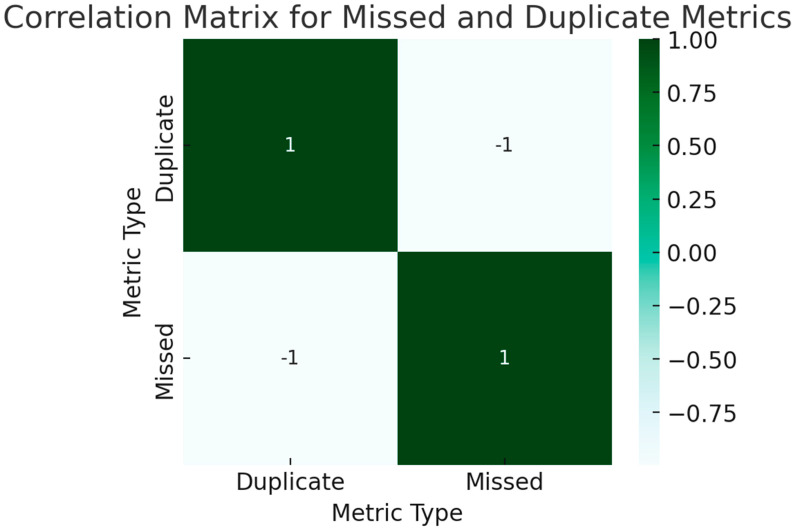
Correlation Matrix for Missed and Duplicate Metrics.

**Table 1 jpm-13-01457-t001:** Validity of provided references from ChatGPT, Bing Chat and Bard AI.

	ChatGPT-3.5 (n = 199)	Bing Chat (n = 158)	Bard (n = 112)	*p*-Value
Accurate	76 (38.2%) *	47 (29.8%) **	3 (2.7%) *,**	<0.001
Inaccurate	82 (41.2%) *	77 (48.7%) **	26 (23.2%) *,**	<0.001
Fabricated	32 (16.1%) *	21 (13.3%) **	71 (63.4%) *,**	<0.001
Incomplete	9 (4.5%)	13 (8.2%)	12 (10.7%)	0.11

* Significant difference between ChatGPT-3.5 and Bard *p* < 0.05. ** Significant difference between Bing Chat and Bard *p* < 0.05.

**Table 2 jpm-13-01457-t002:** Types of inaccuracy detected among the inaccurate references between ChatGPT, Bing Chat and Bard AI.

	ChatGPT-3.5 (n = 82)	Bing Chat (n = 77)	Bard (n = 26)	*p*-Value
Inaccurate DOI	74 (90.3%) *	68 (88.3%) **	18 (69.2%) *,**	0.02
Inaccurate title	4 (4.9%) *	2 (2.6%) **	7 (26.9%) *,**	<0.001
Inaccurate author	18 (22.0%) *	13 (16.9%) **	19 (73.1%) *,**	<0.001
Inaccurate journal/book	10 (12.2%) *	6 (7.8%) **	8 (30.8%) *,**	0.010
Inaccurate year/issue	14 (17.1%) *	7 (9.1%) **	15 (57.7%) *,**	<0.001

* Significant difference between ChatGPT-3.5 and Bard *p* < 0.05. ** Significant difference between Bing Chat and Bard *p* < 0.05.

## Data Availability

Data Availability Statements are available in the original publication, reports, and preprints that were cited in the reference citation.

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
