# Peer review of "Navigating the Landscape of Personalized Medicine: The Relevance of ChatGPT, BingChat, and Bard AI in Nephrology Literature Searches"

_jpm, 2023, doi:10.3390/jpm13101457_

Round 1
Reviewer 1 Report
Dear Authors
The paper titled Navigating the Landscape of Personalized Medicine: The Relevance of ChatGPT, BingChat, and Bard AI in Nephrology Literature Searches aimed to individually assess their citation accuracy within Nephrology, comparing their performance in providing precise.
Authors generated the prompt to solicit 20 references in Vancouver style in each 12 Nephrology topics, using ChatGPT, Bing Chat and Bard. This study's findings emphasize the varied citation accuracy among the AI tools evaluated. Despite some promising results, the discrepancies identified call for a cautious and rigorous vetting of AI-sourced references in medicine. Such chatbots, before becoming standard tools, need substantial refinements to assure unwavering precision in their outputs.
The paper is well conducted, reaching interesting conclusions. Authors should review the English language writing.
Best regards,
Authors should review the English language writing.
Author Response
Reviewer 1
The paper titled Navigating the Landscape of Personalized Medicine: The Relevance of ChatGPT, BingChat, and Bard AI in Nephrology Literature Searches aimed to individually assess their citation accuracy within Nephrology, comparing their performance in providing precise.
Authors generated the prompt to solicit 20 references in Vancouver style in each 12 Nephrology topics, using ChatGPT, Bing Chat and Bard. This study's findings emphasize the varied citation accuracy among the AI tools evaluated. Despite some promising results, the discrepancies identified call for a cautious and rigorous vetting of AI-sourced references in medicine. Such chatbots, before becoming standard tools, need substantial refinements to assure unwavering precision in their outputs.
The paper is well conducted, reaching interesting conclusions. Authors should review the English language writing.
Response: Thank you for your thoughtful review and constructive feedback on our manuscript. We are pleased that you found our research well conducted and reaching interesting conclusion. We have thoroughly addressed the comments you provided.
Comment #1
Authors should review the English language writing.
Response:
We are grateful for your comment concerning the language of the manuscript. As experts in the field, we recognize that the clarity of expression is crucial for conveying complex ideas and research findings effectively. We have carefully revised the areas you highlighted to improve the quality of the English language in our manuscript.
- Original Sentence: The purpose of this study is to assess their citation accuracy of AI models including ChatGPT (versions 3.5 and 4.0) [15], Bing Chat, and Bard AI in retrieving and validating references for academic research in nephrology.
Revised Sentence: The purpose of this study is to assess the citation accuracy of AI models including ChatGPT (versions 3.5 and 4.0) [15], Bing Chat, and Bard AI in retrieving and validating references for academic research in nephrology.
- Original Sentence: We reviewed the references and got rid of duplicated reference from the same AI chatbot before analysis.
Revised Sentence: We reviewed the references and excluded duplicated reference from the same AI chatbot before analysis.
- Original Sentence: The study outcomes delineate distinctive patterns pertaining to citation accuracy within the purview of AI tools.
Revised Sentence: The study outcomes delineate distinctive patterns regarding citation accuracy within the purview of AI tools.
Of note, we changed the word “pertaining to” into “regarding” to avoid redundancy from prior sentence.
- Original Sentence: However, validity of ChatGPT and Bing Chat in providing Nephrology references were not significantly different.
Revised Sentence: However, the validity of ChatGPT and Bing Chat in providing Nephrology references were not significantly different.
We have thoroughly reviewed the entire manuscript and made further adjustments as needed to ensure the quality of English language throughout. We hope that these revisions address your concerns, and we are open to any further suggestions or clarifications.
Once again, thank you for your time and expertise in reviewing our work. We believe that your feedback has significantly enhanced the quality and clarity of our manuscript, and we are grateful for your contribution to this process.

Reviewer 2 Report
Dear Authors,
I congratulate you on your very interesting manuscript.
However, there are some aspects that require your attention.
In the discussion section you need to underline the use of AI in other fields of medical care such as drug prescriptions, drug interactions and allergic reactions. Please reference this to the article by Dumitru M, Berghi ON, Taciuc IA, Vrinceanu D, Manole F, Costache A. Could Artificial Intelligence Prevent Intraoperative Anaphylaxis? Reference Review and Proof of Concept. Medicina (Kaunas). 2022 Oct 26;58(11):1530. doi: 10.3390/medicina58111530. PMID: 36363487; PMCID: PMC9694532. This will help increase the visibility of your article.
Moreover, insert a subsection explaining the limitations of the study.
At the end of the manuscript in the acknowledgement section insert a disclaimer underlining that you did not use the AI systems for writing the article itself.
Reference needs expanding according to reviewer instructions and needs formatting according to MDPI instructions.
Furthermore, insert a graphical abstract of the article.
Looking forward to receiving the improved version of your article.
Author Response
Response to Reviewer#2
I congratulate you on your very interesting manuscript. However, there are some aspects that require your attention.
Comment #1
In the discussion section you need to underline the use of AI in other fields of medical care such as drug prescriptions, drug interactions and allergic reactions. Please reference this to the article by Dumitru M, Berghi ON, Taciuc IA, Vrinceanu D, Manole F, Costache A. Could Artificial Intelligence Prevent Intraoperative Anaphylaxis? Reference Review and Proof of Concept. Medicina (Kaunas). 2022 Oct 26;58(11):1530. doi: 10.3390/medicina58111530. PMID: 36363487; PMCID: PMC9694532. This will help increase the visibility of your article.
Response: We are sincerely grateful for taking your precious time to review our manuscript. Your comprehensive feedback reinforces our motivation to tackle such a pertinent subject in the medical research field. We agree and appreciate your suggestion regarding underlying roles of AI in the other medical fields. We also found the suggested references very helpful and have incorporated in our revised manuscript as new reference# 24. We have added the other examples of AI use, including detection of prescription error, drug interaction, allergic reactions and assistance in specific diet, at the beginning of the discussion part.
Revised Sentence: The advancement of AI within the medical field has led to substantial transformations [20,21], including assisting in specialized diet supports [22,23], prevention of potential allergic reactions [24,25], detection of prescription errors [26], extraction of drug interactions from the literature [27], and particularly concerning literature reviews [28,29].
Comment #2
Moreover, insert a subsection explaining the limitations of the study.
Response: Thank you for your suggestion. We have inserted subsections on the limitation part of our study, including AI platforms, Lack of clinical implications, Limited citation assessment, Variability due to updates, Scope and Validity of databases.
Revised Sentence:
This study bears several limitations that warrant acknowledgment. AI platforms: Firstly, the assessment focused solely on ChatGPT (GPT-3.5 and GPT-4.0), Bing Chat, and Bard AI, excluding other emerging AI platforms that may exhibit distinct citation accuracy profiles. Lack of clinical implications: Secondly, the study did not investigate the impact of reference inaccuracies on downstream research, clinical decision-making, or patient outcomes, which could provide crucial insights into the practical implications of AI-generated references in the medical domain. Limited citation assessment: While the study accounted for discrepancies in citation elements such as DOIs and author names, it did not delve into potential errors in other bibliographic components such as correctness of Vancouver format or page ranges. This omission could underestimate the full scope of inaccuracies present in AI-generated references. Variability due to updates: Furthermore, the study's reliance on AI models for reference generation introduces the possibility of variability in performance due to updates or modifications in the models' algorithms over time. The investigation was conducted with specific versions of AI models, and as these models undergo continuous refinement, their citation accuracy may evolve. Scope: Additionally, the study's sample size of Nephrology topics and AI-generated references might not fully capture the breadth of medical literature or the complexity of citation accuracy in other medical specialties. The study's exclusive focus on AI chatbots limits the exploration of potential variations in citation accuracy among different AI-powered tools, such as summarization algorithms or natural language processing applications. Validity of databases: Finally, the assessment of AI-generated references relied on cross-referencing with established databases, assuming the accuracy of these databases. Any errors or discrepancies present in the reference databases could influence the study's findings and conclusions.
Comment #3
At the end of the manuscript in the acknowledgement section insert a disclaimer underlining that you did not use the AI systems for writing the article itself.
Response: Thank you for your suggestion. We acknowledge the need for clarification that AI platforms were not employed for writing. We have attested and clarified our writing in the Acknowledgments as suggested.
Acknowledgments: The manuscript presents a research study that employed AI chatbots for its investigations. Notably, it incorporated the use of ChatGPT versions GPT-3.5 and GPT-4.0, both developed by OpenAI. Additionally, the study made use of Bing Chat, a product of Microsoft, and Bard AI, which is based on the PaLM architecture and is a Google initiative. The manuscript also features a flow diagram crafted using Whimsical. The other contents of this manuscript were generated by the authors and did not employ AI systems for writing purposes. All content written in this manuscript results from the authors’ contributions.
Comment #4
Reference needs expanding according to reviewer instructions and needs formatting according to MDPI instructions.
Response: We appreciate your attention to detail. We have inserted relevant references according to the reviewer’s guidance and formatted the references into MDPI style.
Comment #5
Furthermore, insert a graphical abstract of the article.
Response:
We are grateful for your remark on the graphical abstract. We agree and thus have generated a graphical abstract of the article as suggested.
We hope that these revisions address your concerns, and we are open to any further suggestions or clarifications. Again, thank you for your invaluable contribution to the enhancement of our research through your review.
Thank you for your time and consideration. We greatly appreciated the reviewer's and editor's time and comments to improve our manuscript. The manuscript has been improved considerably by the suggested revisions.
